# Buckling of Cracked Euler–Bernoulli Columns Embedded in a Winkler Elastic Medium

José Antonio Loya *[ID], Carlos Santiuste [ID], Josué Aranda-Ruiz [ID] and Ramón Zaera [ID]

Department of Continuum Mechanics and Structural Analysis, University Carlos III of Madrid, Avda. de la Universidad, 30, 28911 Leganés, Madrid, Spain; csantius@ing.uc3m.es (C.S.); jaranda@ing.uc3m.es (J.A.-R.); rzaera@ing.uc3m.es (R.Z.)
* Correspondence: jloya@ing.uc3m.es

**Abstract:** This work analyses the buckling behaviour of cracked Euler–Bernoulli columns immersed in a Winkler elastic medium, obtaining their buckling loads. For this purpose, the beam is modelled as two segments connected in the cracked section by a mass-less rotational spring. Its rotation is proportional to the bending moment transmitted through the cracked section, considering the discontinuity of the rotation due to bending. The differential equations for the buckling behaviour are solved by applying the corresponding boundary conditions, as well as the compatibility and jump conditions of the cracked section. The proposed methodology allows calculating the buckling load as a function of the type of support, the parameter defining the elastic soil, the crack position and the initial length of the crack. The results obtained are compared with those published by other authors in works that deal with the problem in a partial way, showing the interaction and importance of the parameters considered in the buckling loads of the system.

**Keywords:** buckling loads; cracked Euler–Bernoulli columns; Winkler elastic medium





## 1. Introduction

The stability analysis of beam-type elements subjected to compressive loads is crucial for the design of structures in the civil, mechanical, aerospace, nuclear and offshore fields. Buckling is one of the most common modes of instability in columnar structures, as well as in elements supported by the ground, such as pipes, piles, footings, or railway tracks. Due to their geometric characteristics, these structures are often modelled as beams in civil engineering or railway engineering, with the Euler–Bernoulli or Timoshenko theory being the most commonly used. Additionally, when considering the effect exerted by the soil on the beam, the Winkler and the Pasternak model (or a combination of both) are frequently employed for elastic medium, where the soil is treated as a set of uniformly distributed linear springs without or with shear interaction between them. Thus, a beam-soil system can be analysed using the classical buckling theory, including in the governing equation one or two additional terms [1,2].

In the scientific literature of the last decade, one can find numerous papers dealing with the problem of beam buckling in elastic media. Some focus on the development of mathematical problem-solving techniques, while others focus on the treatment of beams with specific characteristics. In the first group are the works of Aristizabal-Ochoa [3], which deal with the treatment of generalised boundary conditions, that of Hassan and Hadima [4], which obtains the solution using the recursive differentiation method, the work of Anghel and Mares [5], which proposes an integral formulation for an elastic medium of the Winkler–Pasternak type, and that of Ike [6], which employs the Stodola–Vianello iterative method. In the second group, we find the work of Soltani for a tapered Timoshenko beam [7], that of Mohammed et al. [8], which considers FG Euler–Bernoulli beams supported on Winkler–Pasternak elastic foundation, the work of Nguyen et al. [9], that of Mellal et al. [10] for higher-order porous FG beams resting on variable elastic foundations, or the work by Xu

et al. [11] for Timoshenko nanobeams resting on Winkler–Pasternak foundations submitted to thermal loads.

On the other hand, the presence of cracks in these types of structures can lead to a decrease in stiffness, the magnitude of which depends primarily on the element's geometry, support conditions, crack size and crack position. This decrease in stiffness has a significant effect on the beam's buckling load. The treatment of the crack involves representing the damaged section with a linear rotational spring, whose stiffness is related to the crack size and the geometry of the cross-section. Continuity in displacement, bending moment, and shear force are imposed on the cracked section, as well as a discontinuity in the rotation that is proportional to the transmitted bending moment. Using this methodology (as well as other equivalent alternatives that consider flexural stiffness singularities using the Dirac delta function), the problem of buckling in cracked beams has been studied in other works [12–16]. Recently, this methodology has also been applied to the analysis of buckling instabilities in FG beams [17] and in nanocantilevers [18].

In the works cited above, either the beam embedded in the elastic medium is not flawed, or the presence of a crack but not the effect of the surrounding elastic medium is taken into account. In the first case, the models do not allow consideration of the flexibil-ising effect of an eventual crack in the structural element, which leads to overestimates of the buckling load. In the second case, the models do not include the stabilising effect of the elastic medium, resulting in an oversizing of the cracked structural element. The joint consideration of the crack and the elastic medium has been taken into account for the vibratory analysis of beams [19], but not for the study of buckling instability. There-fore, the present study analyses the case of slender Euler–Bernoulli columns with cracks immersed in an elastic medium subjected to compression, considering the discontinuity of bending motion in that section. The effect of the crack length and position, stiffness of the surrounding medium, and type of supports on the determination of critical buckling loads are examined.

## 2. Euler–Benoulli Column Model in an Elastic Medium

### 2.1. Theoretical Formulation of an Intact Column

Consider a column without cracks, with length $L$, width $B$, height $W$, cross-sectional moment of inertia $I$, Young's modulus $E$, immersed in an elastic medium of Winkler type with stiffness $k_w$, and subjected to compression, as shown in Figure 1.

Following the classical Euler's theory, the equation governing the buckling behaviour of a column with a uniform cross-section embedded in an elastic medium can be pre-sented as:

$$EI\frac{d^4v(x)}{dx^4} + P_c\frac{d^2v(x)}{dx^2} + k_wv(x) = 0 \tag{1}$$

where $v(x)$ is the transverse deflection of the column, $x$ is the longitudinal Cartesian coordinate of the column with the origin at the bottom support, and $P_c$ is the critical buckling load.

Considering the following dimensionless variables:

$$\xi = \frac{x}{L}, \quad V = \frac{v}{L}, \quad \lambda^2 = \frac{P_cL^2}{EI}, \quad K_w = \frac{k_wL^4}{EI} \tag{2}$$

Equation (1) can be rewritten as:

$$V(\xi)^{iv} + \lambda^2 V(\xi)'' + K_w V(\xi) = 0 \tag{3}$$

where $(\cdot)'$ represents the derivative with respect to $\xi$. The general solution for the equation with constant coefficients (3) can be expressed using hyperbolic functions. However, considering the definitions of rotation $\theta$ and stresses (bending moment, $M$, and shear force,

*Q*), the corresponding nondimensional variables (rotation, bending moment, $\overline{M}$, and shear force $\overline{Q}$) can be written from their dimensional counterparts:

$$\theta = V', \quad \overline{M} = \frac{ML}{EI} = V'', \quad \overline{Q} = \frac{QL^2}{EI} = V''' + \lambda^2 V' \tag{4}$$

So, the solution of Equation (3) can be conveniently expressed in terms of the nondimensional displacement, slope, bending moment, and shear force at $\xi = 0$, ($V_0$, $\theta_0$, $\overline{M}_0$, $\overline{Q}_0$), and Krylov–Duncan functions, $g_i(\xi)$ [20]:

$$V(\xi) = V_0 \cdot g_1(\xi) + \theta_0 \cdot g_2(\xi) + \overline{M}_0 \cdot g_3(\xi) + \overline{Q}_0 \cdot g_4(\xi) \tag{5}$$

where the following functions and parameters are defined:

$$g_1(\xi) = \cosh(\Lambda_1 \xi) - \frac{\Lambda_1^2 \cosh(\Lambda_1 \xi)}{(\Lambda_1^2 - \Lambda_2^2)} - \frac{\Lambda_1^2 \Lambda_2^2 \cosh(\Lambda_2 \xi)}{(\Lambda_1^2 - \Lambda_2^2)} \tag{6}$$

$$g_2(\xi) = \frac{\sinh(\Lambda_1 \xi)}{\Lambda_1} - \frac{(\lambda^2 + \Lambda_1^2)\sinh(\Lambda_1 \xi)}{\Lambda_1(\Lambda_1^2 - \Lambda_2^2)} + \frac{(\lambda^2 + \Lambda_1^2)\sinh(\Lambda_2 \xi)}{\Lambda_2(\Lambda_1^2 - \Lambda_2^2)} \tag{7}$$

$$g_3(\xi) = \frac{\cosh(\Lambda_1 \xi)}{(\Lambda_1^2 - \Lambda_2^2)} - \frac{\cosh(\Lambda_2 \xi)}{(\Lambda_1^2 - \Lambda_2^2)} \tag{8}$$

$$g_4(\xi) = -\frac{\sinh(\Lambda_1 \xi)}{\Lambda_1(\Lambda_1^2 - \Lambda_2^2)} + \frac{\sinh(\Lambda_2 \xi)}{\Lambda_2(\Lambda_1^2 - \Lambda_2^2)} \tag{9}$$

with

$$\Lambda_{1,2} = \sqrt{\frac{-\lambda^2 \pm \sqrt{\lambda^4 - 4K_w}}{2}} \tag{10}$$

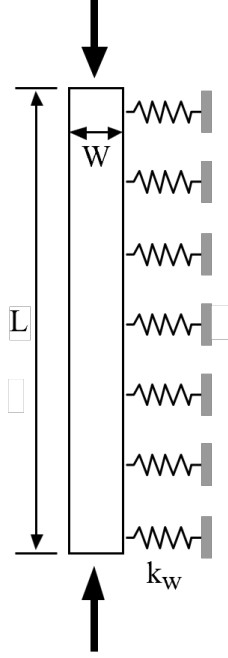

**Figure 1.** Euler–Bernoulli column in an elastic medium of Winkler type.

In order to solve it, the corresponding boundary conditions must be applied at each end (Table 1).

**Table 1.** Boundary Conditions.

| Type of Support | Restrictions |
|---|---|
| Simple support | $V = \overline{M} = 0$ |
| Fixed support | $V = \theta = 0$ |
| Free end | $\overline{M} = \overline{Q} = 0$ |

### 2.2. Problem Formulation in Cracked Columns

Consider an open crack of length *a* located at a distance $\beta = (b/L)$ from the lower support, as sketched in Figure 2 (left). Following the method proposed by Freund and Herrmann in 1976 [21] and followed by many other authors [22–27], the cracked column is considered as two segments connected by elastic rotational springs, as shown in Figure 2 (right), whose stiffness depends on the crack depth and the geometry of the cracked section.

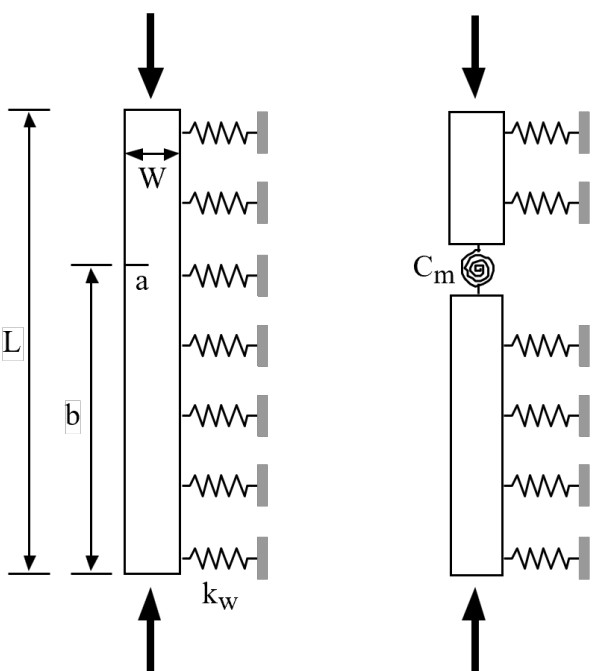

**Figure 2.** (**Left**): Cracked Euler–Bernoulli column in an elastic medium of Winkler type. (**Right**): Equivalent model with elastic spring for the cracked section.

The discontinuity in the deflection slope of the column at the cracked section, $\Delta\theta$, is proportional to the bending moment transmitted by that section, $M(b)$, as follows:

$$\Delta\theta = C_m M(x)|_{x=b} \tag{11}$$

$$C_m = \frac{W}{EI} m(\alpha, \text{cross-section geometry}) \tag{12}$$

with $\alpha = (a/W)$ being the dimensionless crack length and the function *m* evaluated by Tada et al. in 1985 [28] using the theory of Fracture Mechanics, which can be applied to linear elastic material behaviour. For the specific case of a rectangular cross-section beam, the function *m* takes the form expressed in Equation (13) [28]:

$$m(\alpha) = 2\left(\frac{\alpha}{1-\alpha}\right)^2 (5.93 - 19.69\alpha + 34.14\alpha^2 - 35.84\alpha^3 + 13.2\alpha^4) \tag{13}$$

Based on Equation (3), the governing equation for each of the two segments can be expressed as follows:

$$V_1(\xi)^{iv} + \lambda^2 V_1(\xi)'' + K_w V_1(\xi) = 0; \quad 0 < \xi < \beta \tag{14}$$

$$V_2(\xi)^{iv} + \lambda^2 V_2(\xi)'' + K_w V_2(\xi) = 0; \quad \beta < \xi < 1 \tag{15}$$

where $\lambda$ is an eigenvalue of the cracked column in the elastic medium, related to the critical buckling load of the column with a crack-through expression (2).

The above equations must be solved by applying the corresponding boundary conditions, as well as the following continuity and jump conditions at the cracked section, $\xi = \beta$. These conditions are collected in the following Equations (16)–(19):

- Continuity in deflection:

$$\Delta V = V_2(\beta) - V_1(\beta) = 0 \tag{16}$$

- Continuity in bending moment:

$$V_2''(\beta) = V_1''(\beta) \tag{17}$$

- Continuity in shear force:

$$V_2'''(\beta) + \lambda^2 V_2'(\beta) = V_1'''(\beta) + \lambda^2 V_1'(\beta) \tag{18}$$

- Jump in the slope deflection:

$$\Delta\theta = V_2'(\beta) - V_1'(\beta) = \eta V_2''(\beta) \tag{19}$$

where $\eta = \frac{W}{L} m(\alpha) V_2$ is the parameter that controls the severity of the crack.

## 3. Direct Solution

The direct solution for the buckling forces can be obtained by separately analysing the column segments on each side of the crack. The solution which satisfies the compatibility conditions at the crack for each segment, $V_i(\xi)$, $i = 1, 2$, can be expressed in terms of displacements, rotations, and forces at the bottom support, $\xi = 0$, ($V_0$, $\theta_0$, $\overline{M}_0$, $\overline{Q}_0$, respectively), as well as of the discontinuity in rotation, $\Delta\theta$, at the cracked section, $\xi = \beta$:

$$V_1(\xi) = V_0 \cdot g_1(\xi) + \theta_0 \cdot g_2(\xi) + \overline{M}_0 \cdot g_3(\xi) + \overline{Q}_0 \cdot g_4(\xi) \tag{20}$$
$$V_2(\xi) = V_1(\xi) + \Delta\theta \cdot g_2(\xi - \beta) \tag{21}$$

Applying the boundary conditions at the ends and the compatibility conditions at the cracked section (automatically satisfied by the expressions (20) and (21)), the displacement functions $V_{1,2}(\xi)$ lead to the corresponding eigenvalue problem, whose solution allows for obtaining the critical buckling loads of the system.

## 4. Numerical Results

### 4.1. Influence of the Crack on the Buckling Load

Firstly, the proposed model has been applied to Euler–Bernoulli columns without cracks and in the absence of an elastic medium ($k_w = 0$), with different boundary conditions (simply supported, clamped-clamped, clamped-free, clamped-supported). The obtained eigenvalues and the corresponding critical buckling loads, $P_c$, show perfect agreement with the cases described by other authors [16].

Subsequently, the influence of the severity of the crack and the position of the cracked section has been analysed in the absence of an elastic medium. The obtained critical loads for cracks with dimensionless length ($a/W$) ranging from 0 to 0.9, and located at a distance $\beta = 0.25$ and $\beta = 0.5$ from the bottom support, coincide with those calculated in other

studies [16]. These results are normalised with respect to the value corresponding to the intact case and are represented in Figures 3 and 4, respectively.

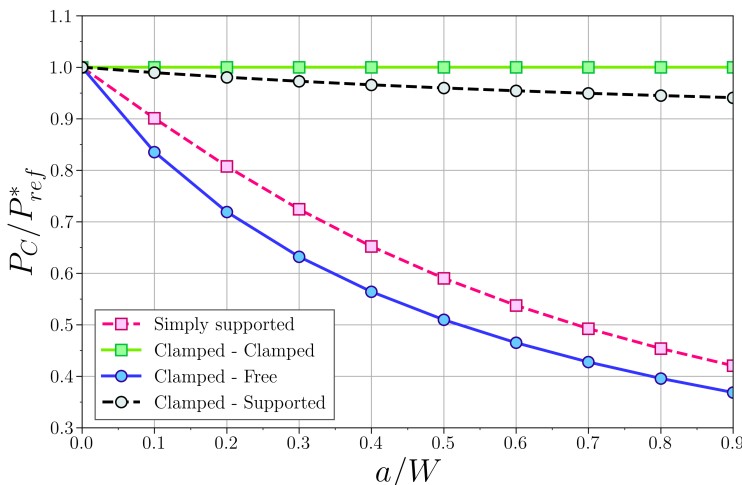

**Figure 3.** Variation of the first normalised critical buckling load with ($a/W$) for different boundary conditions and $\beta = 0.25$.

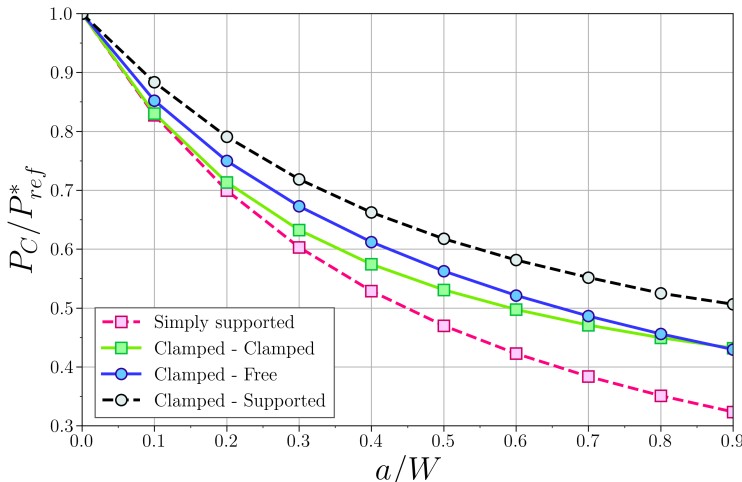

**Figure 4.** Variation of the first normalised critical buckling load with ($a/W$) for different boundary conditions and $\beta = 0.50$.

As expected, it is observed that the buckling load decreases with crack length. Regarding the position of the cracked section, it affects significantly. As a representative case, the clamped–clamped case can be observed: when the crack section is located at $\beta = 0.25$ (Figure 3), the crack has no effect on the first buckling load due to the null bending moment at the cracked section, in agreement with other works [16]. In the clamped–supported case, the bending moment in the cracked section at $\beta = 0.25$ (Figure 3) is small but not null, so the effect of the crack on the first buckling load is limited. These behaviours differ greatly in the case of $\beta = 0.50$ (Figure 4).

### 4.2. Influence of the Crack and Elastic Medium on the Buckling Load

To analyse the applicability of the model when the column is embedded in an elastic medium, the case of a simply supported intact steel column with a length $L = 1$ m, a moment of inertia $I = 833.333$ mm$^4$, and a Young's modulus $E = 200$ GPa [29] is considered. The values of the critical buckling load, $P_c$, obtained for different values of the dimensionless parameter representative of the Winkler medium, $K_w$ [0, 5, 10, 50, 100],

along with those calculated by Jančo in 2013 [29] using the analytical solution proposed for the simply supported column (Equation (22)), are presented in Table 2:

$$P_c = \frac{(K_w + \pi^4)EI}{\pi^2 L^2} \tag{22}$$

**Table 2.** Comparison of results for simply supported column without crack in an elastic medium.

| $P_c$ [N] | $K_w$ | | | | |
|---|---|---|---|---|---|
| | 0 | 5 | 10 | 50 | 100 |
| Theoretical [29] | 1644.94 | 1729.36 | 1814.82 | 2489.28 | 3334.65 |
| Proposed | 1644.94 | 1729.44 | 1813.94 | 2489.95 | 3334.97 |
| Error [%] | 0 | 0 | 0.01 | 0.03 | 0.04 |

The comparison between the theoretical results [29] and the calculated ones for the simply supported case showed a good correlation between solutions. The maximum error was below 0.04%, validating the proposed solution for the elastic medium.

Additionally, Figure 5 shows the variation of the normalised critical load with the stiffness of the medium for other boundary conditions. In all cases, an increase in the critical load with $K_w$ was observed, which was more pronounced as the boundary conditions become less rigid, as in the case of simply supported and cantilever beams.

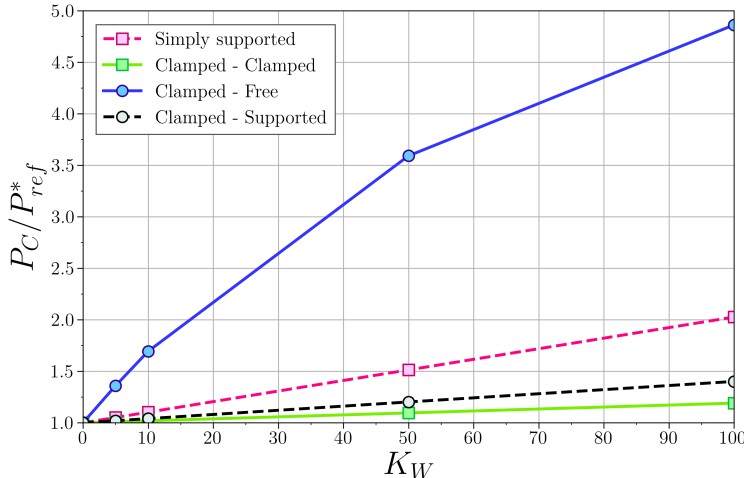

**Figure 5.** Variation of the first normalised critical buckling load with $K_w$ and different boundary conditions for a non-cracked beam.

### 4.3. Combined Influence of the Crack and the Elastic Medium on the Buckling Load

In this section, the coupled effect of the cracked section, both at $\beta = 0.25$ and $\beta = 0.50$, and the elastic medium for different boundary conditions (simply supported, clamped–pinned, clamped–clamped, and cantilever) are analysed in Figures 6–13, respectively, representing the variation of the critical buckling load with the crack severity factor $a/W$ and the Winkler stiffness $K_w$. In all cases, the load was normalised with that of an intact column in a surrounding elastic medium with zero stiffness, for the same boundary conditions.

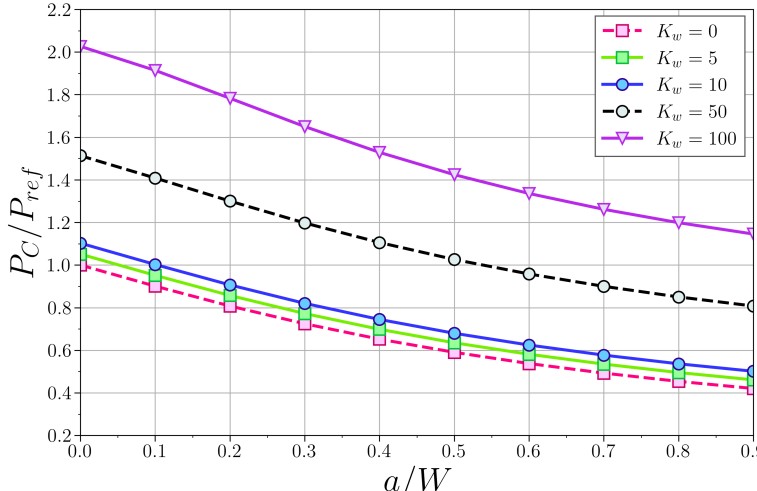

**Figure 6.** Simply supported column, $\beta = 0.25$. Variation of the first normalised critical buckling load with $(a/W)$ and for different $K_w$.

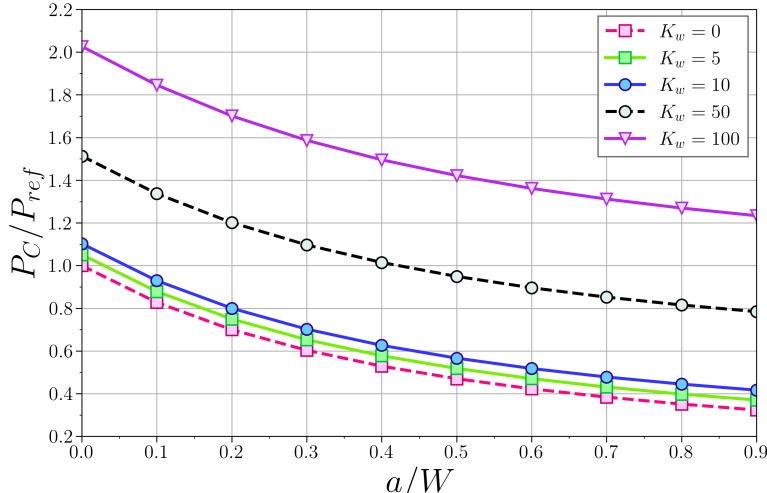

**Figure 7.** Simply supported column, $\beta = 0.5$. Variation of the first normalised critical buckling load with $(a/W)$ and for different $K_w$.

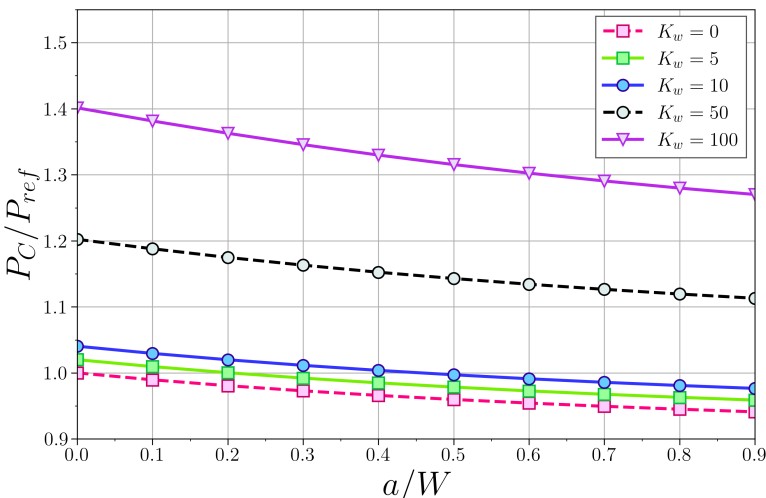

**Figure 8.** Clamped–supported column, $\beta = 0.25$. Variation of the first normalised critical buckling load with $(a/W)$ and for different $K_w$.

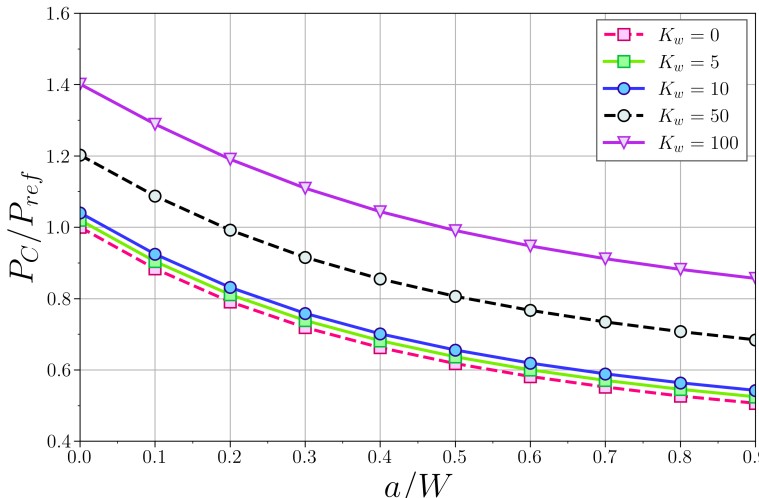

**Figure 9.** Clamped–supported column, $\beta = 0.5$. Variation of the first normalised critical buckling load with ($a/W$) and for different $K_w$.

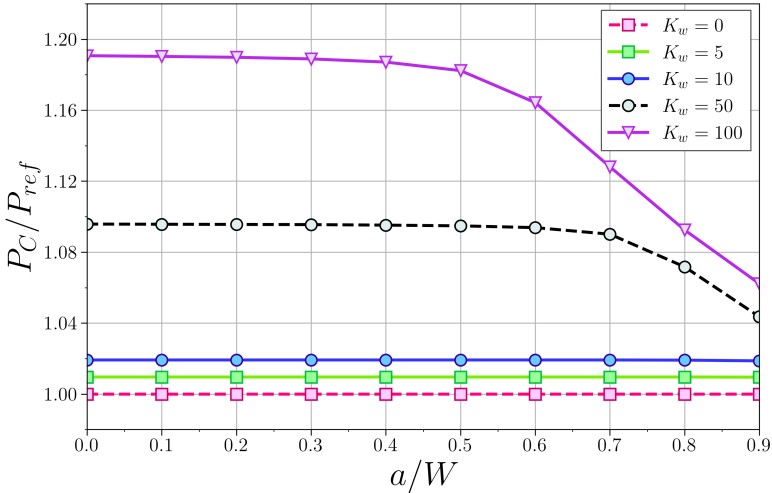

**Figure 10.** Clamped–clamped column, $\beta = 0.25$. Variation of the first normalised critical buckling load with ($a/W$) and for different $K_w$.

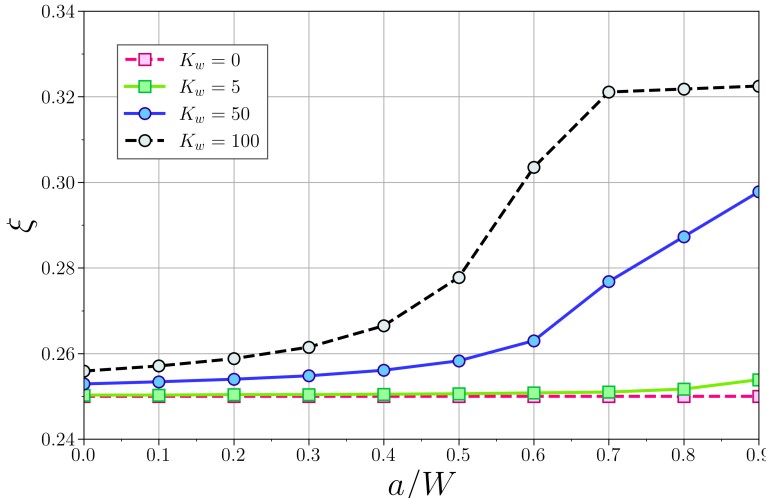

**Figure 11.** Position $\xi$ at which the bending moment becomes null in a clamped–clamped column, for different crack lengths ($a/W$) and $K_w$.

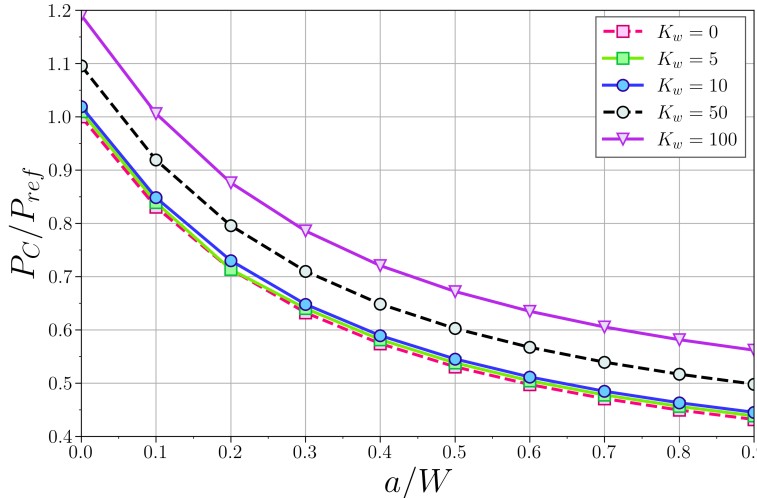

**Figure 12.** Clamped–clamped column, $\beta = 0.5$. Variation of the first normalised critical buckling load with $(a/W)$ and for different $K_w$.

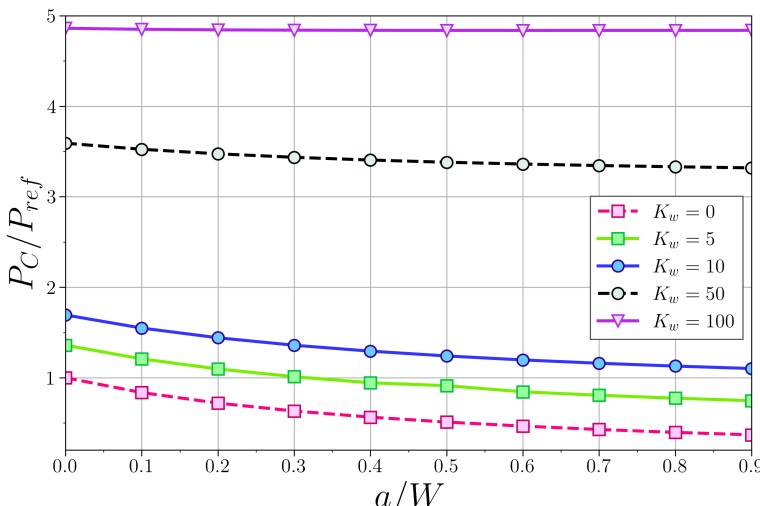

**Figure 13.** Cantilever column, $\beta = 0.25$. Variation of the first normalised critical buckling load with $(a/W)$ and for different $K_w$.

### 4.3.1. Simply Supported Column

The first boundary conditions analysed were for simply supported columns, see Figures 6 and 7. In these cases, a decrease in the critical buckling load with the crack length was observed, while, on the other hand, the load increased with the stiffness of the medium due to the higher restriction to the column displacement. Although the trends were similar in all cases, the percentage decrease in the buckling load with crack severity was smaller as $K_w$ increased due to the higher transverse stiffness of the system.

In general, these conclusions are also applicable to the other boundary conditions (clamped–clamped, clamped–pinned, and cantilever), with some particularities, as discussed below.

### 4.3.2. Clamped–Supported Column

The clamped–supported column behaviour is presented in Figures 8 and 9. Firstly, the influence of the elastic medium was weaker for these boundary conditions, as compared to the simply supported case, showing smaller increases in the buckling load as $K_w$ rose (an average increase of 50% is observed compared to an increase of 170% in the case of simply

supported column). Therefore, the potential benefit of the elastic confinement provided by the Winkler medium was reduced in the embedded–supported beam.

### 4.3.3. Clamped–Clamped Column

For the clamped–clamped column, and for the crack-position $\beta = 0.25$ (Figure 10), it is worth noting that the crack severity hardly affected the buckling load for low values of $K_w$. However, for large values, its influence begins to be noticeable for cracks larger than half the section height $W$.

Figure 11 represents the distance from the lower support, $\xi$, where the bending moment, $M$, was null in a clamped–clamped column with a cracked section at $\beta = 0.25$. Notice that for $Kw = 0$, $\overline{M} = 0$ at the cracked position for any crack severity, so no jump in slope deflection was achieved (Equation (19)). If the stiffness of the elastic medium increases, the position where $\overline{M} = 0$ does too, even in the case of an intact column. The contribution of the crack severity allows this position to increase from $\xi = 0.25$ up to $\xi = 0.32$ (close to $1/3$ length of the column). This implies that the bending moment at the cracked section is now non-nil, leading to a jump in the slope deflection in the cracked section.

For $\beta = 0.50$, see Figure 12, similar behaviour as in the simply supported case was appreciated, showing the importance of the crack position in the buckling behaviour for this boundary condition, which can strongly alter the trends.

### 4.3.4. Cantilever Column

The clamped-free (or cantilever) column leads to the lowest stiffness among all the boundary conditions considered. Here, the stabilising effect of the elastic medium was very relevant, as can be seen in Figures 13 and 14. Here, the buckling load increased by 500% in the whole range of Winkler stiffnesses, as compared to a scarce 20% in the clamped–clamped case (that with the higher structural stiffness and buckling load). This underlines the importance of the stiffness of the Winkler medium in the safety of cracked structures, especially for those with higher flexibility.

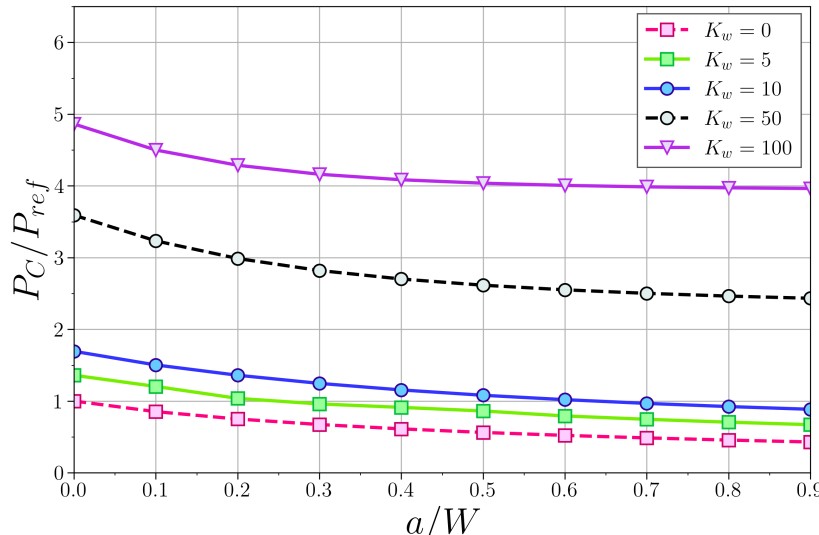

**Figure 14.** Cantilever column, $\beta = 0.5$. Variation of the first normalised critical buckling load with $(a/W)$ and for different $K_w$.

## 5. Conclusions

It is known that the presence of cracks in a compression-loaded column reduces the buckling forces of the structure by making it more flexible. In this work, a theoretical model is proposed to determine these critical loads in cracked Euler–Bernoulli columns immersed in a Winkler-type elastic medium, under different boundary conditions. The described

method divides the column into two segments connected by a rotational elastic massless spring, whose stiffness is related to the bending moment transmitted at the cracked section, and satisfies the corresponding continuity and jump conditions at the crack.

The proposed model has been independently validated against literature results for both the case of cracked columns in the absence of an elastic medium and intact columns immersed in a Winkler-type elastic medium.

Furthermore, the coupled effect of crack presence and the elastic medium was analysed, revealing the opposing effect on the critical buckling load produced by both considerations.

The present model has shown its capability to reproduce the buckling behaviour of a cracked beam in an elastic medium. However, some simplifying hypotheses were assumed and they can be modified in future works to obtain a better understanding of their limitations. For instance, the contribution of the shear forces can be added using a Timoshenko beam and/or modelling the elastic foundation with a Pasternak model.

**Author Contributions:** Conceptualisation, J.A.L. and R.Z.; methodology, J.A.L.; formal analysis, J.A.L. and R.Z.; investigation, J.A.L., C.S., J.A.-R. and R.Z.; writing—original draft preparation, J.A.L., C.S., J.A.-R. and R.Z.; writing—review and editing, J.A.L., C.S., J.A.-R. and R.Z.; supervision, J.A.L. and R.Z.; funding acquisition, C.S. and J.A.L. All authors have read and agreed to the published version of the manuscript.

**Funding:** This research was funded by the Spanish State Research Agency, grant number PID2020-118946RB-I00.

**Conflicts of Interest:** The authors declare no conflict of interest.

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
