# Peer review of "Buckling of Cracked Euler–Bernoulli Columns Embedded in a Winkler Elastic Medium"

_mca, doi:10.3390/mca28040087_

Round 1

Reviewer 1 Report

The buckling of structure is very important point to be investigated. However, the following article present a fundamental analysis of cracked Euler-Bernoulli beam with  elastic foundation that has been considered previously by many many articles. For example :- 

Caddemi, S., & Calio, I. (2008). Exact solution of the multi-cracked Euler–Bernoulli column. International Journal of Solids and Structures45(5), 1332-1351.

Mohammed, A. T., Hareb, M. A., & Eqal, A. K. (2021, February). Investigation on the Analysis of Bending and Buckling for FGM Euler-Bernoulli Beam Resting on Winkler-Pasternak Elastic Foundation. In Journal of Physics: Conference Series (Vol. 1773, No. 1, p. 012027). IOP Publishing.   (this paper include FGM that is missed in the following article)

and others. 

In addition, authors must be review the latest articles available on website. The most recent reference include in this proposal was published 2013 , from 10 years ago. 

Therefor, I must reject the article.  

Author Response

Reviewer 1:

The buckling of structure is very important point to be investigated. However, the following article present a fundamental analysis of cracked Euler-Bernoulli beam with elastic foundation that has been considered previously by many many articles. For example :

  • Caddemi, S., & Calio, I. (2008). Exact solution of the multi-cracked Euler–Bernoulli column. International Journal of Solids and Structures, 45(5), 1332-1351.
  • Mohammed, A. T., Hareb, M. A., & Eqal, A. K. (2021, February). Investigation on the Analysis of Bending and Buckling for FGM Euler-Bernoulli Beam Resting on Winkler-Pasternak Elastic Foundation. In Journal of Physics: Conference Series(Vol. 1773, No. 1, p. 012027). IOP Publishing. (this paper include FGM that is missed in the following article) and others. 

In addition, authors must be review the latest articles available on website. The most recent reference include in this proposal was published 2013 , from 10 years ago.

Answer to Reviewer 1:

The authors are grateful for the reviewer's valuable comments, and have carried out a thorough literature search where they have indeed been able to find additional previous works, some of them very recent, dealing with the buckling problem of a beam embedded in an elastic medium, or with the buckling problem of a beam with one or several cracks. All these papers have been cited and commented in the new version of the introduction.

However, the quoted papers consider either the presence of a crack in the beam without elastic medium, or the effect of the elastic medium in non-cracked beams, but not both effects. This justifies the interest of the present work, which studies the stabilizing of the effect of the elastic medium on the buckling load to avoid potential oversizing of the cracked structural element.

Reviewer 2 Report

In this study authors have analysed the response of an Euler beam which is resting on an elastic foundation and subjected to a compressive force. It is also assumed that a crack is present in the body of the beam. The governing equation of the beam is established when elastic medium is of the Winkler tpye. The governing equation is established in dfimensionless presentatio and after that is solved exactly. Different types of edge support such as clamped, simply supported and free are considered. The crack reduction stiffness is modeled by a rotational spring. The solution method is straightforward and exact.  The manuscript is well-written and well-organised. It provides a step by step procedure which may be of interest for the future readers. The manuscript may be accepted for publication when the following comments are considered correctly

1) Why the elastic foundation is modelled via the simple Winkler model? At least the shear stiffness of the foundation known as the Pasternak foundation may be considered. Why it is ignored

2) In Eq. (4) it is obvious that bending moment and shear force are in dimensionless form. So please provide the definition of these parameters in dimensionless form

3) More presentation about Eq. (13) should be given. How this equation is obtained? Experimentally or theoretically? Does it depends on the material type?

4) As seen from Figure (3) the effect of crack depth is ignorable for C-C boundary conditions. Why? Can authors explain?

5) It is known that elastoc foundation may change the buckling mode of S-S beams. So the formula (22) may not be valid always and is valid only for low values of foundation parameter. Have authors considered this fact in their results?

Author Response

Reviewer2

In this study authors have analysed the response of an Euler beam which is resting on an elastic foundation and subjected to a compressive force. It is also assumed that a crack is present in the body of the beam. The governing equation of the beam is established when elastic medium is of the Winkler tpye. The governing equation is established in dfimensionless presentatio and after that is solved exactly. Different types of edge support such as clamped, simply supported and free are considered. The crack reduction stiffness is modeled by a rotational spring. The solution method is straightforward and exact.  The manuscript is well-written and well-organised. It provides a step by step procedure which may be of interest for the future readers. The manuscript may be accepted for publication when the following comments are considered correctly

1) Why the elastic foundation is modelled via the simple Winkler model? At least the shear stiffness of the foundation known as the Pasternak foundation may be considered. Why it is ignored

Authors thank the pertinent reviewer's recommendation. Effectively, the elastic foundation can be modelled using different approaches as Winkler model, Pasternak model or a combination of both. In this work, the elastic foundation was modeled using the Winkler model for simplification. To the authors' knowledge, this is the first analysis of the buckling behavior of Euler-Bernoulli cracked beams in an elastic medium, and such a model can provide information for a better understanding of their behavior.

According to this point, Lines 216-220 has been included:

“The present model has shown its capability to reproduce the buckling behavior of a cracked beam in an elastic medium. However, some simplifying hypotheses were assumed and they can be modified in future works to get a better understanding of their limitations. For instance, the contribution of the shear forces can be added using a Timoshenko beam and/or modelling the elastic foundation with Pasternak model.”

2) In Eq. (4) it is obvious that bending moment and shear force are in dimensionless form. So please provide the definition of these parameters in dimensionless form

Authors thank the pertinent reviewer's recommendation. Therefore, L78-81, and equation (4) are rewritten:

“However, considering the definitions of rotation θ and stresses (bending moment, M, and shear force, Q), the corresponding nondimensional variables (rotation, bending moment, M, and shear force Q) can be written from their dimensional counterparts:”

3) More presentation about Eq. (13) should be given. How this equation is obtained? Experimentally or theoretically? Does it depends on the material type?

This expression is obtained by Tada & Paris in 1985 applying the theory of the fracture mechanics, and can be applied for linear elastic material behavior.

Lines 97-100 of the manuscript have been rewritten:

“…the function m evaluated by Tada et al. in 1985 [31] using the theory of Fracture Mechanics, that can be applied to linear elastic material behaviour. For the specific case of a rectangular cross-section beam, the function m takes the form expressed in Equation 13 [31]:”

4) As seen from Figure (3) the effect of crack depth is ignorable for C-C boundary conditions. Why? Can authors explain?

Authors thank the pertinent reviewer's question. Lines 140-145 of the manuscript has been rewritten:

“…when the crack section is located at β = 0.25 (Figure 3), the crack has no effect on the first buckling load due to the null bending moment at the cracked section, in agreement with other works [16]. In the clamped-supported case, the bending moment in the cracked section at β = 0.25 (Figure 3) is small but no null, so the effect of the crack on the first buckling load is limited. These behaviours differ greatly in the case of β = 0.50 (Figure 4).”

Later in the document, new Fig.11 is included. It represents the position where the bending moment is null in a clamped-clamped column with cracked section at β = 0.25. When Kw=0, it is null just at the cracked position, independently of the crack severity. Therefore, there is no jump in the slope deflection (Eq.19) in this case. Meanwhile, the bending nil moment position moves from β = 0.25 when the Winkler stiffness and the crack-length increases (Fig.12) up to β = 0.32 (close 1/3 length of the column). From the point of view of the cracked section, this change means that the bending moment at β = 0.25 is no null and the presence of the crack leads to a jump in the slope deflection.

5) It is known that elastic foundation may change the buckling mode of S-S beams. So the formula (22) may not be valid always and is valid only for low values of foundation parameter. Have authors considered this fact in their results?

Thank you for the question. In this work we intended to independently validate the proposed model, on the one hand, the inclusion of the crack and, on the other hand, the effect of the surrounding elastic medium.

In this particular section of the work, the inclusion of Winkler's model is validated with the results presented by Janˇco 2013 [16] for a simply supported beam without cracking in a Winkler elastic medium. As the reviewer indicates, the formula proposed for the theorical values (Eq.5 [33]) is applicable only for such boundary conditions, so this is the case considered for validation.

Applying the dimensionless analysis proposed in the work under review (Eq.4) to the stiffness range considered by Janco (k=0-1E6 N/m2), the corresponding dimensionless Kw are obtained (see attached table).

k (Janco 2013 [33])

0

100

1000

10000

100000

1000000

Kw [33] (by Eq.4)

0

1

6

60

600

6000

As can be observed, the Kw values used in the evaluated work (Kw=0,1,5,10,50,100) are within the study range of Janˇco [16]. The comparison between the numerical results obtained in the present work with those obtained by means of the Janˇco dimensionless formula (Eq.22) show a high correlation (see Table 2 of the manuscript).

Lines 150-153 of the manuscript has been rewritten:

“The values of the critical buckling load, Pc, obtained for different values of the dimensionless parameter representative of the Winkler medium, Kw [0, 5, 10, 50, 100], along with those calculated by Janco in 2013 [ 17] using the analytical solution proposed for the simply supported column (Equation 22), are presented in Table 2:”

Reviewer 3 Report

The manuscript analyses the buckling behaviour of cracked Euler-Bernoulli columns immersed in a Winkler elastic medium, obtaining buckling loads. For this purpose, the beam is modelled as two segments connected in the cracked section by a mass-less rotational spring. Its rotation is proportional to the bending moment transmitted through the cracked section, considering the discontinuity of the rotation due to bending. The manuscript contains some merits, and the content is interesting. However, some major problems must be carefully addressed and explained for reconsideration:

1.      The presentation structure in the introduction to lead the research gap and discuss the new contribution is somehow unclear and still messed up. The authors are hence advised to revise further the introduction section to present deficiencies or shortcomings of other studies to make a bridge or research gap to introduce the novelty of their work instead of a long introduction paragraph.

2.      The limitations of the proposed method and future research work should be mentioned in the manuscript.

3.      I do not find any latest research papers in the reference. All the cited articles are from 2013 or before. Why?

4.      How did the authors decide the location of the crack? Is there any specific reason, or is it just a random location?

5.      The results section should be concise and pointwise, highlighting the unique outcome, not the general one.

6.      The authors should include more detail in Figures 3 and 4. Also, specify the reason behind the behaviour of the curve, specifically about the clamped-clamped and clamped-supported conditions in Figures 3 and 4.

7.     In Figures 10 and 11, if everything in the figure is the same except the beta value, then why is the curve response so different? Explain.

Author Response

[MCA] Manuscript ID: mca-2508337– Answer to reviewer 3

Reviewer 3:

The manuscript analyses the buckling behaviour of cracked Euler-Bernoulli columns immersed in a Winkler elastic medium, obtaining buckling loads. For this purpose, the beam is modelled as two segments connected in the cracked section by a mass-less rotational spring. Its rotation is proportional to the bending moment transmitted through the cracked section, considering the discontinuity of the rotation due to bending. The manuscript contains some merits, and the content is interesting. However, some major problems must be carefully addressed and explained for reconsideration:

1 The presentation structure in the introduction to lead the research gap and discuss the new contribution is somehow unclear and still messed up. The authors are hence advised to revise further the introduction section to present deficiencies or shortcomings of other studies to make a bridge or research gap to introduce the novelty of their work instead of a long introduction paragraph.

The authors are grateful for the reviewer's comment. The introduction section has been modified, extending on the one hand the analysis of previous articles related to the work presented and, on the other hand, highlighting that these articles do not jointly consider the effect of the elastic medium and the presence of cracks in the structural element, which may lead to overestimation or underestimation of the buckling load in a design problem.

2 The limitations of the proposed method and future research work should be mentioned in the manuscript.

Authors thank the pertinent reviewer's recommendation. The next paragraph was added to the conclusions section:

“The present model has shown its capability to reproduce the buckling behavior of a cracked beam in an elastic medium. However, some simplifying hypotheses were assumed and they can be modified in future works to get a better understanding of their limitations. For instance, the contribution of the shear forces can be added using a Timoshenko beam and/or modelling the elastic foundation with Pasternak model.”

  1. I do not find any latest research papers in the reference. All the cited articles are from 2013 or before. Why?

The authors are grateful for the reviewer's valuable comments, and have carried out a thorough literature search where they have indeed been able to find additional previous works, some of them very recent, dealing with the buckling problem of a beam embedded in an elastic medium, or with the buckling problem of a beam with one or several cracks. All these papers have been cited and commented in the new version of the introduction. However, the quoted papers consider either the presence of a crack in the beam without elastic medium, or the effect of the elastic medium in non-cracked beams, but not both effects.

  1. How did the authors decide the location of the crack? Is there any specific reason, or is it just a random location?

The specific crack locations have been selected according to other papers, as [16], to validate the crack effect consideration in the model presented.

  1. The results section should be concise and pointwise, highlighting the unique outcome, not the general one.

Thank you very much for encourage us to improve our outcome in this section. Authors made a general review, rewritten and extended some explanation regarding to the results obtained. Some of them, that need higher attention as the reviewer detected in questions 6 and 7, are included:

Lines 140-145 of the manuscript has been rewritten for clarification:

“…when the crack section is located at β = 0.25 (Figure 3), the crack has no effect on the first buckling load due to the null bending moment at the cracked section, in agreement with other works [16]. In the clamped-supported case, the bending moment in the cracked section at β = 0.25 (Figure 3) is small but no null, so the effect of the crack on the first buckling load is limited. These behaviours differ greatly in the case of β = 0.50 (Figure 4).”

Lines 191-202 of the manuscript and Figure 11 have been included:

“Figure 11 represents the distance from the lower support, ξ, where the bending moment, M, is null in a clamped-clamped column with a cracked section at β = 0.25. Notice that for Kw = 0, M = 0 is just at the cracked position for any crack severity, so no jump in the slope deflection is achieved (equation (19)). If the stiffness of the elastic medium increases, the M = 0 position does, even in the case of an intact column. The contribution of the crack severity allows this position to increase from ξ = 0.25 up to ξ = 0.32 (close 1/3 length of the column). This implies that the bending moment at the cracked section is now non-nil, leading to a jump in the slope deflection in the cracked section. Figure 11. Position ξ at which the bending moment becomes null in a clamped-clamped column, for different crack lengths (a/W) and Kw. For β = 0.50, see Figure 12, similar behaviour as in the simply supported case is appreciated, showing the importance of the crack position in the buckling behaviour for this boundary condition, which can strongly alter the trends.”

  1. The authors should include more detail in Figures 3 and 4. Also, specify the reason behind the behaviour of the curve, specifically about the clamped-clamped and clamped-supported conditions in Figures 3 and 4.

Author thank the reviewer to highlight this point. Lines 140-145 of the manuscript has been rewritten for clarification:

“…when the crack section is located at β = 0.25 (Figure 3), the crack has no effect on the first buckling load due to the null bending moment at the cracked section, in agreement with agree with other works [16]. In the clamped-supported case, the bending moment in the cracked section at β = 0.25 (Figure 3) is small but no null, so the effect of the crack on the first buckling load is limited. These behaviours differ greatly in the case of β = 0.50 (Figure 4).”

  1. In Figures 10 and 11, if everything in the figure is the same except the beta value, then why is the curve response so different? Explain.

This pertinent question is connected to the previous one. A new figure (Fig.11) is included in the document (attached also below) representing the distance from the lower support where the bending moment is null in a clamped-clamped column with the cracked section at β = 0.25.

When Kw=0 (related to the previous question), the bending moment is null just at the cracked position, independently of the crack severity. Therefore, there is no jump in the slope deflection for any crack length (Eq.19). Meanwhile, the position where the bending moment is null moves from β = 0.25 when the Winkler stiffness and the crack-length increases (Fig.11) up to β = 0.32 (close 1/3 length of the column). This implies that bending moment at the cracked section is now non-nil, leading to a jump in the slope deflection.

Lines 191-202 of the manuscript and Figure 11 have been included:

“Figure 11 represents the distance from the lower support, ξ, where the bending moment, M, is null in a clamped-clamped column with a cracked section at β = 0.25. Notice that for Kw = 0, M = 0 is just at the cracked position for any crack severity, so no jump in the slope deflection is achieved (equation (19)). If the stiffness of the elastic medium increases, the M = 0 position does, even in the case of an intact column. The contribution of the crack severity allows this position to increase from ξ = 0.25 up to ξ = 0.32 (close 1/3 length of the column). This implies that the bending moment at the cracked section is now non-nil, leading to a jump in the slope deflection in the cracked section. Figure 11. Position ξ at which the bending moment becomes null in a clamped-clamped column, for different crack lengths (a/W) and Kw. For β = 0.50, see Figure 12, similar behaviour as in the simply supported case is appreciated, showing the importance of the crack position in the buckling behaviour for this boundary condition, which can strongly alter the trends.”

Round 2

Reviewer 1 Report

The authors considered all comments. 

Reviewer 2 Report

The revised version is satisfactory and it may be accepted in present form

Reviewer 3 Report

The revised manuscript is accepted for publication.